# Engineering the pore environment of antiparallel stacked covalent organic frameworks for capture of iodine pollutants

Yinghui Xie[1], Qiuyu Rong[1], Fengyi Mao[1], Shiyu Wang[1], You Wu[1], Xiaolu Liu[1], Mengjie Hao[1], Zhongshan Chen[1], Hui Yang [1] ✉, Geoffrey I. N. Waterhouse [2], Shengqian Ma [3] ✉ & Xiangke Wang [1] ✉

Radioiodine capture from nuclear fuel waste and contaminated water sources is of enormous environmental importance, but remains technically challenging. Herein, we demonstrate robust covalent organic frameworks (COFs) with antiparallel stacked structures, excellent radiation resistance, and high binding affinities toward $I_2$, $CH_3I$, and $I_3^-$ under various conditions. A neutral framework (ACOF-1) achieves a high affinity through the cooperative functions of pyridine-N and hydrazine groups from antiparallel stacking layers, resulting in a high capacity of ~2.16 g/g for $I_2$ and ~0.74 g/g for $CH_3I$ at 25 °C under dynamic adsorption conditions. Subsequently, post-synthetic methylation of ACOF-1 converted pyridine-N sites to cationic pyridinium moieties, yielding a cationic framework (namely ACOF-1R) with enhanced capacity for triiodide ion capture from contaminated water. ACOF-1R can rapidly decontaminate iodine polluted groundwater to drinking levels with a high uptake capacity of ~4.46 g/g established through column breakthrough tests. The cooperative functions of specific binding moieties make ACOF-1 and ACOF-1R promising adsorbents for radioiodine pollutants treatment under practical conditions.

Nuclear power represents a low-carbon energy source[1], thereby offering an important transitional technology in the shift away from polluting fossil fuel energy[2,3]. However, nuclear fission reactors produce toxic radionuclides capable of causing immeasurable harm if released into the environment through accidents or improper nuclear waste processing, disposal or storage[4–6]. Among the potentially troublesome radionuclides are the isotopes of iodine that account for ~0.69% of the $^{235}U$ slow-neutron fission products, including short-lived $^{131}I$ and long-lived $^{129}I$, which typically exist as $I_2$ and $CH_3I$ in air, or molecular/ionic species in water ($I_2 + I^- \rightleftharpoons I_3^-$)[4,7]. Due to the highly volatile nature of $I_2$ and $CH_3I$, these molecules are readily released into air during the acid dissolution step used in spent fuel reprocessing[8]. Additionally, molecular/ionic iodine species exist in acidic nuclear waste[8]. Due to their radiotoxicity, volatility at relatively lower

temperatures, and high mobility, radioactive $I_2$, $CH_3I$, and $I_3^-$ can quickly spread into the environment, impacting living organisms and inducing disease[7,9–12]. In addition, nuclear accidents caused by human activity can also lead to the release of large amounts of radioactive iodine[7]. Effective radioactive iodine management, particularly at the first stage of the spent fuel reprocessing process, is essential for the safe disposal of radioactive waste from power generation and the prevention of off-site migration of contaminated aqueous solutions[13,14]. However, radioactive iodine management and environmental remediation are technically very challenging, owing to the complexity of the systems harboring the iodine species.

To address the challenge of radioiodine capture, researchers seek adsorbents that can selectively capture iodine species to allow long-term storage and mitigate environmental risks[15]. Various materials

[1]College of Environmental Science and Engineering, North China Electric Power University, Beijing 102206, P.R. China. [2]School of Chemical Sciences, The University of Auckland, Auckland 1142, New Zealand. [3]Department of Chemistry, University of North Texas, Denton, TX 76201, USA. ✉e-mail: h.yang@ncepu.edu.cn; shengqian.ma@unt.edu; xkwang@ncepu.edu.cn

such as silver-based nanostructures[16,17], zeolites[18], aerogels[19,20], layered double hydroxides (LDHs)[21], polymers[22–24], porous aromatic frameworks (PAFs)[25], organic compounds and cages[26–31], metal oxo clusters[32,33], metal-organic frameworks (MOFs)[34–41], hydrogen-bonded organic frameworks (HOFs)[42], and covalent organic frameworks (COFs)[43–53] have been explored as adsorbents for iodine capture. Common strategies for achieving high performance iodine adsorbents include (i) trapping $I_2$ with electron-deficient silver-containing compounds to produce AgI precipitates[16]; (ii) preparing large surface area adsorbents with cage-like structures to achieve high $I_2$ uptake capacities[54]; (iii) preparing adsorbents supporting electron-rich N, S, or O heteroatoms to improve the interaction with $I_2$ through charge transfer effects[23,42]; (iv) construction of pyridinium-N moieties for trapping $CH_3I$ through an N-methylation reaction[51]; (v) incorporating ionic moieties in adsorbents for coulombic interactions with $I_3^-$[31]. These approaches rely on optimization of the binding affinity between adsorbent and specific iodine species, along with tuning the spatial environment and exposed binding sites of adsorbents. However, inorganic materials demonstrate relatively low capacity towards $I_2$, $CH_3I$, and $I_3^-$. Most porous inorganic materials spontaneously release adsorbed iodine because of the weak interactions between the pore of framework and iodine molecules. Moreover, most of the adsorption studies reported to date have focused on the adsorptive-capture of $I_2$ vapor, $CH_3I$, and/or $I_3^-$ in closed systems, with only a few works exploring dynamic iodine species removal. Moreover, since radioactive molecular iodine and organic iodides (e.g., $CH_3I$) coexist in off-gas streams and $I_3^-$ dissolved in contaminated water, it is particularly important to develop adsorbents that can dynamically and efficiently capture these diverse iodine-containing molecules/or ions under practical conditions, motivating detailed investigations.

Discovering strategies to improve the host-guest interaction between porous materials and iodine species is the key to overcoming the aforementioned challenges. With these considerations in mind, we hypothesized that by converging cooperative functions into the nanospace of COFs, it should be possible to achieve control over the recognition and dynamic uptake efficiency of different iodine species. Ideally, adsorbents are needed which can capture $I_2$ and $CH_3I$ during

the first stage of spent fuel reprocessing (i.e. from spent fuel dissolved in nitric acid solutions), as well as $I_3^-$ capture from wastewater containing radioactive iodine ions. To achieve this ambitious task, we herein selected a COF supporting electron-rich pyridine-N linkers and nearby hydrazine sites, possessing an antiparallel stacking π-electron conjugated framework (denoted as ACOF-1), as a proof-of-concept adsorbent for iodine species (Fig. 1a). The pyridine-N groups efficiently bind $CH_3I$ and $I_2$ molecules, with the binding affinity being improved by nearby hydrazine sites. Moreover, the antiparallel stacked layers in ACOF-1 form three-dimensional "multi-N nanotrap sites", significantly enhancing the utilization and affinity of chelating sites. Subsequently, post-synthetic methylation of the pyridine-N sites created cationic pyridinium-N units, with the obtained COF (denoted as ACOF-1R) offering a high density of accessible $I_3^-$ exchange sites. Through detailed experimental studies, ACOF-1 was shown to offer an exceptional affinity toward $CH_3I$ under various conditions, achieving a high dynamic uptake performance of ~0.74 g/g at 25 °C, making it one of the best $CH_3I$ extractants reported to date. In addition, a packed column containing cationic ACOF-1R showed exceptional adsorption properties toward aqueous $I_3^-$ (with a high uptake capacity of ~4.46 g/g), rapidly decontaminating $I_3^-$ polluted groundwater to drinking water level. The excellent performance of the developed ACOF-1 and ACOF-1R, together with their applicability in diverse settings, confirm their promise as adsorbents for iodine pollutants under practical conditions, such as from nuclear waste and contaminated water.

## Results
### Synthesis and characterization
To address the challenge of dynamically capturing different iodine pollutants under various conditions, unique antiparallel stacked COFs (ACOF-1 and ACOF-1R) were synthesized. The synthetic strategy is illustrated in Fig. 1b. We reacted 5,5',5''-(benzene-1,3,5-triyl)tripicolinaldehyde (BTPA) and 2,5-dibutoxyterephthalohydrazide (DBTH) in a mixture of mesitylene/dioxane/AcOH (5:5:1, by volume) at 120 °C for 3 days. After filtration and washing with ethanol and acetone, a pale yellow powder was obtained (ACOF-1). We subsequently investigated the structural components of ACOF-1 using various spectroscopic

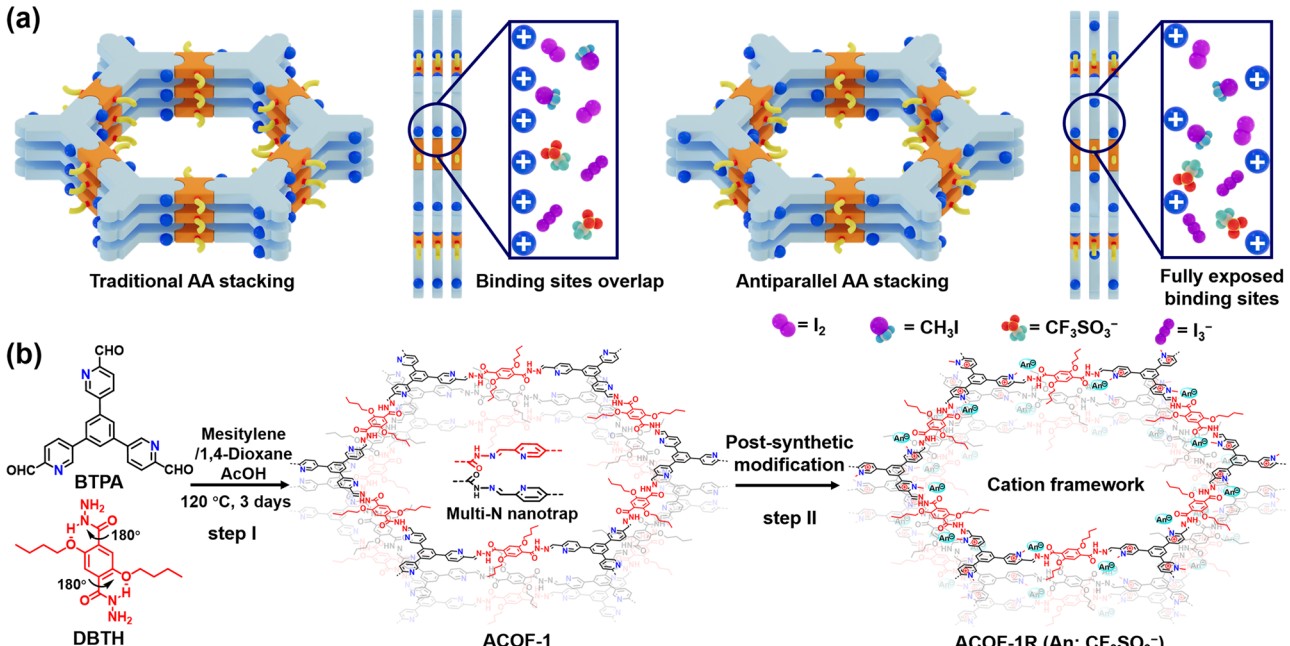

**Fig. 1 | Schematic illustration of the stacking model and synthesis of ACOF-1 and ACOF-1R. a** Schematic illustration of traditional AA stacking (left) and antiparallel AA stacking (right) in COF adsorbents, showing how antiparallel stacked layers form three-dimensional "multi-N nanotraps", significantly enhancing the utilization and affinity of chelating sites. **b** Illustration of the synthesis of ACOF-1 and ACOF-1R.

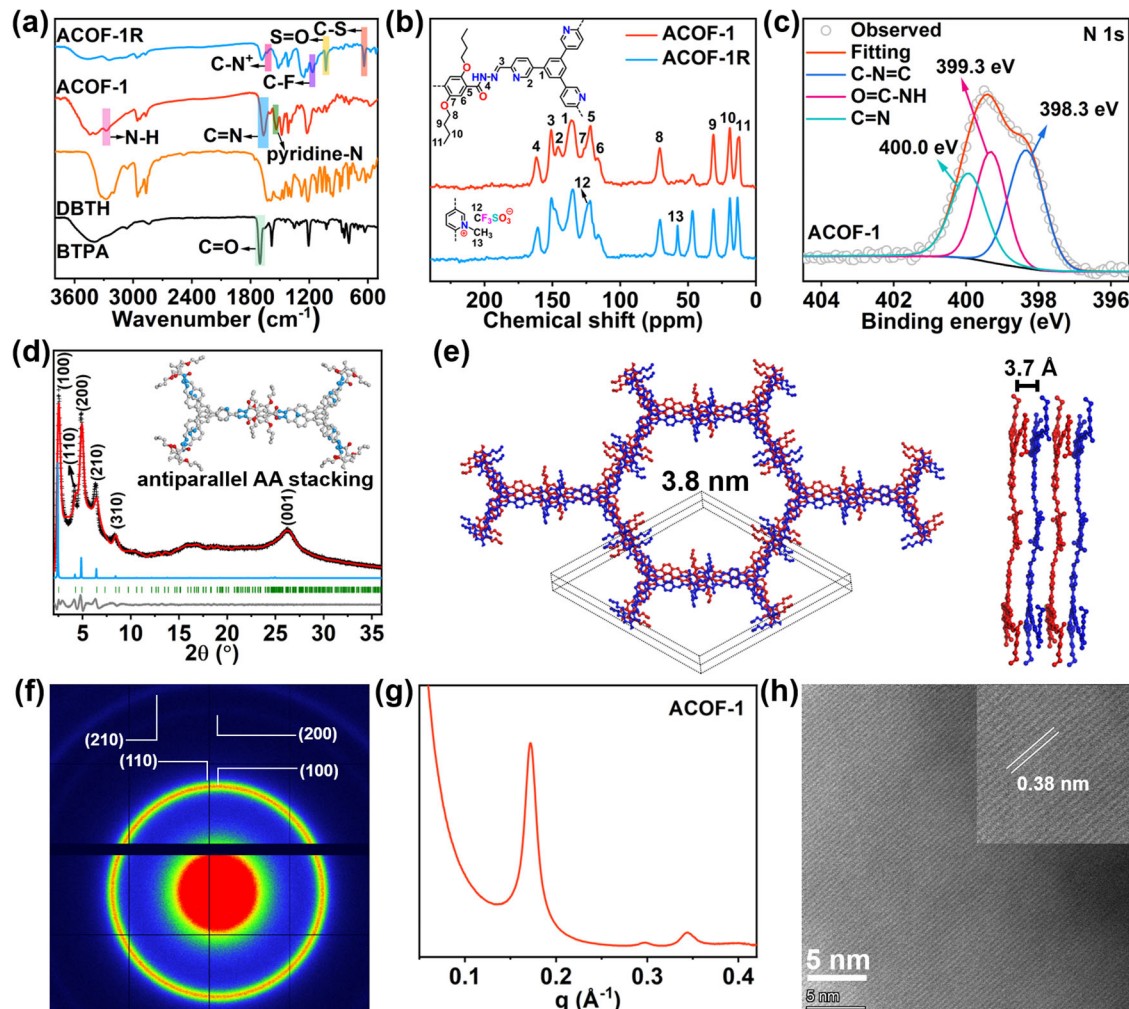

**Fig. 2 | Chemical structure and characterization of ACOFs. a** FT-IR spectra of ACOF-1, ACOF-1R, and linkers. **b** $^{13}$C CP-MAS solid-state NMR spectra of ACOF-1 and ACOF-1R. **c** N 1 s XPS spectrum of ACOF-1. **d** Experimental PXRD patterns of ACOF-1 with corresponding Pawley refinement (red), simulated results (blue), and Bragg positions (green) showing good fits to the experimental data (black) with minimal differences (gray). The inset shows the structural model of ACOF-1 assuming the antiparallel AA stacking mode. **e** Top and side view of the antiparallel AA stacking crystal structure of ACOF-1. **f, g** 2D SAXS image and pattern of ACOF−1. **h** HRTEM image of ACOF-1 (inset: expanded view showing the interlayer distance).

techniques. The Fourier-transform infrared (FT-IR) spectrum of ACOF-1 showed an intense peak at 1668 cm$^{-1}$ (Fig. 2a). This peak will contain contributions from C=N (imine) and C=O stretching vibrations in ACOF-1, with the imine groups being formed through condensation reactions between aldehyde and hydrazine groups in BTPA and DBTH, respectively[55,56]. The presence of imine bonds was further confirmed by the $^{13}$C cross-polarization magic angle spinning (CP-MAS) NMR spectrum of ACOF-1, which showed a characteristic imine resonance signal at 150.8 ppm (Fig. 2b). Further signals at 70.8, 31.1, 18.9, and 12.7 ppm could be attributed to the butoxy group in the DBTH component. The N 1 s X-ray photoelectron spectroscopy (XPS) spectrum of ACOF-1 was deconvoluted into three peaks at 398.3, 399.3, and 400.0 eV, which could readily be assigned to C−N=C (i.e., pyridinic-N), O=C−NH, and C=N, respectively. The data suggested the successful construction of a COF with the desired structure (Fig. 2c, Supplementary Fig. 1).

We next performed powder X-ray diffraction (PXRD) measurement and structural modeling in Materials Studio software to verify that the synthesized COF possessed the designed structure. The experimental PXRD pattern of ACOF-1 showed peaks at 2θ angles of ~2.5°, 4.2°, 4.9°, 6.5°, and 8.4°, which were assigned to the (100), (110), (200), (210), and (310), respectively, of the COF (Fig. 2d). Pawley

refinement revealed good consistency with the experimentally measured PXRD pattern, revealing an antiparallel AA stacking geometry with negligible residuals (R$_p$, 3.89% and R$_{wp}$, 5.71%). The findings confirmed the correctness of the proposed crystal structure (Supplementary Table 1). ACOF-1 crystallizes in a hexagonal P6cc space group with lattice parameters of a = b = 42.31 Å, c = 7.23 Å, α = β = 90°, γ = 120° (Supplementary Table 1). On the basis of these results, ACOF-1 adopted an antiparallel AA stacking mode to form a two-dimensional (2D) structure with a theoretical dynamic pore size of 3.8 nm (Fig. 2e, Supplementary Fig. 2). The 2D layers utilized hydrogen bonding and electrostatic interactions, with an interlayer distance is approximately 3.7 Å (Fig. 2e). For further probe the stacking structure, we carried out small and wide-angle X-ray scattering (SAXS/WAXS) measurements to corroborate the successful synthesis of ACOF-1 (Fig. 2f, g, Supplementary Fig. 3). The (100), (110), (200), and (210) planes of ACOF-1 were observed by 2D SAXS/WAXS images and patterns, consistent with the PXRD and simulated antiparallel AA stacking results. The high-resolution transmission electron microscopy (HRTEM) image of ACOF-1 revealed lattice fringes with spacing around 0.38 nm, close to the calculated interlayer distance in the AA structure, verifying π-π stacking (Fig. 2h). The scanning electron microscopy (SEM) image of ACOF-1 showed a

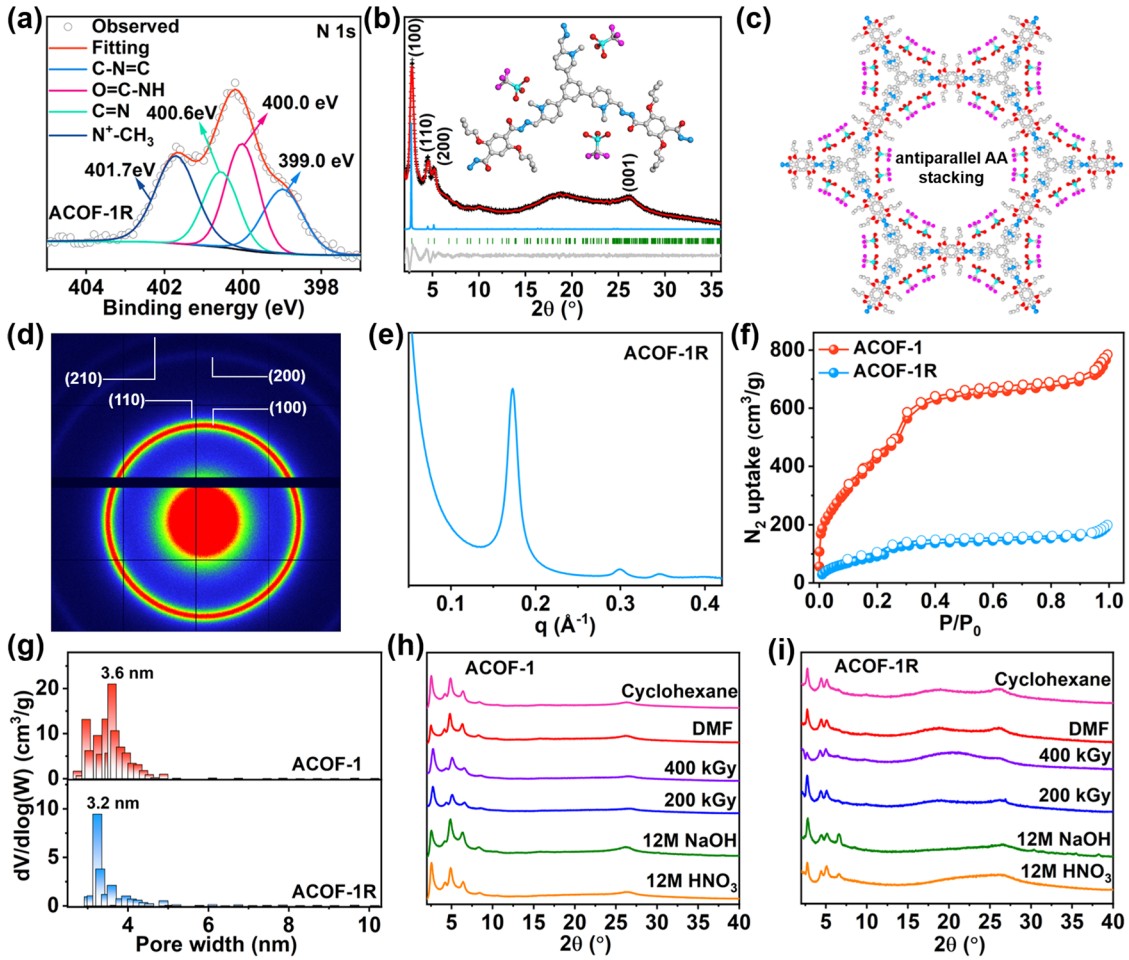

**Fig. 3 | Structural, porosity, and stability of ACOFs. a** N 1 s XPS spectrum of ACOF-1R. **b** Experimental PXRD patterns of ACOF-1R with corresponding Pawley refinement (red), simulated results (blue), and Bragg positions (green) showing good fits to the experimental data (black) with minimal differences (gray). **c** Top view of the antiparallel AA stacking crystal structure of ACOF-1R. **d, e** 2D SAXS image and pattern of ACOF-1R. **f, g** N$_2$ sorption isotherms and pore size distributions measured at 77 K for ACOF-1 and ACOF-1R, respectively. **h, i** PXRD patterns of ACOF-1 and ACOF-1R after treatment under different conditions.

lychee-like nodular morphology (Supplementary Fig. 4). In addition, we selected the DBTH linker for the COF synthesis due to the intramolecular hydrogen bonding involving the DBTH linker (N-H···O), which plays a pivotal role in orienting the hydrazine groups on the linkers, thereby causing ACOF-1 to crystallize in the antiparallel AA stacking mode. This was further revealed by comparison of the charge density map and interlayer differential charge density of antiparallel AA stacking and eclipsed AA stacking structures (Supplementary Fig. 5, Supplementary Table 4). The electrostatic repulsion of alkyl oxygen/hydrazine groups increases the interlayer distance and decreases the interlayer interaction in the eclipsed AA stacking structure. However, the antiparallel AA stacking mode avoids charge repulsion due to intra- and interlayer hydrogen bonds and electrostatic interactions, thus promoting the formation of the antiparallel AA stacking structures. Taken together, we concluded that the ACOF-1 adopted the structural model in Fig. 1b.

To further expand the aforementioned protocol, we next reacted ACOF-1 with methyl trifluoromethanesulfonate to produce a cationic framework (denoted as ACOF-1R) through methylation (Fig. 1b, step II)[57]. The appearance of C−N$^+$ (1630 cm$^{-1}$), C−S (639 cm$^{-1}$), S=O (1030 cm$^{-1}$), and C−F (1166 cm$^{-1}$) stretches in FT-IR spectrum of ACOF-1R revealed the successful chemical transformation of the pyridine groups (in ACOF-1) into pyridine-N$^+$ groups (in ACOF-1R), with CF$_3$SO$_3^-$ ions located as a guest close to the inner pore wall (Figs. 1b, 2a). Solid-state $^{13}$C NMR analyses further confirmed this transformation,

evidenced by the emergence of a C−N$^+$ signal at 57.7 ppm (Fig. 2b). XPS spectra confirmed the presence of pyridine-N$^+$ sites and CF$_3$SO$_3^-$ anions after the methylation reaction (Fig. 3a, Supplementary Fig. 1). PXRD and its simulated results revealed the crystallinity of the COF was retained after the methylation reaction (Fig. 3b, Supplementary Table 2). As expected, the CF$_3$SO$_3^-$ anions are located close to the pyridine-N$^+$ sites (Fig. 3b, c). ACOF-1R maintained a P6cc space group, with slightly changed unit cell parameters (a = b = 42.14 Å and c = 7.38 Å) compared to ACOF-1 (a = b = 42.31 Å, c = 7.23 Å). Moreover, 2D SAXS/WAXS image and pattern showed the intensity of (100), (110), (200), and (210) reflections in ACOF-1R (from q = 0.2-2.0 Å$^{-1}$) were slightly reduced compared with the ACOF-1, suggesting the pore environment was changed after methylation (Fig. 3d, e, Supplementary Fig. 6). SEM and TEM images revealed ACOF-1R retained the lychee-like morphology of ACOF-1 (Supplementary Fig. 7). Elemental analysis showed the N, C, H, and S contents in ACOF-1R were 8.32%, 44.96%, 3.81%, and 7.59%, close to the theoretically calculated results (Supplementary Table 3). HAADF-STEM and mapping images for C, N, O, F, and S showed that all elements were uniformly dispersed in ACOF-1R (Supplementary Fig. 8).

The porosities of the ACOF-1 and ACOF-1R were determined by nitrogen adsorption-desorption isotherms collected at 77 K. The sorption isotherms exhibited a type-II adsorption profile, featuring sharp uptake at P/P$_0$ < 0.05, followed by a further steep region of uptake at P/P$_0$ from 0.05 to 0.35, suggesting the existence of

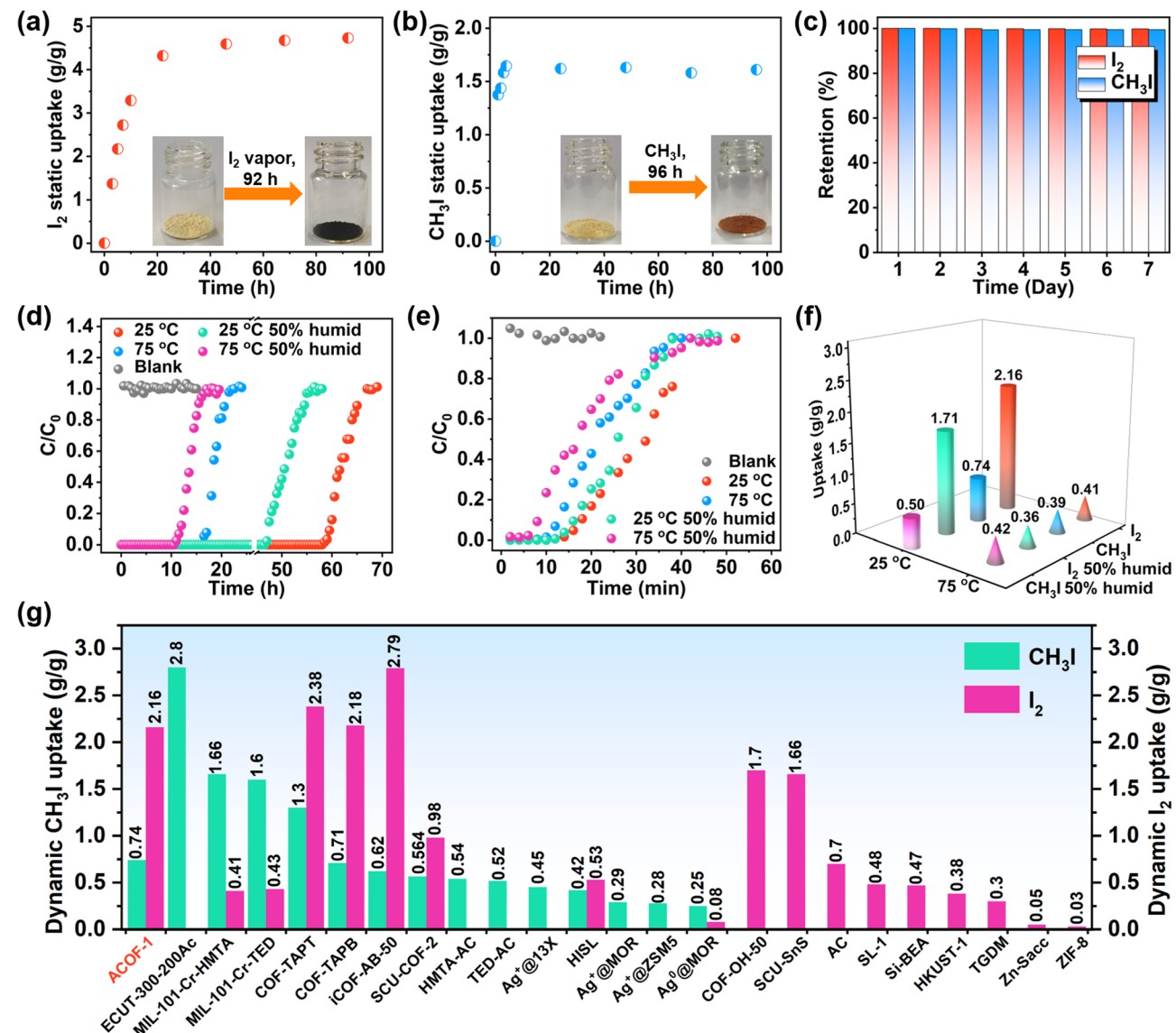

**Fig. 4 | Static and dynamic I$_2$ and CH$_3$I adsorption performance. a** Static I$_2$ vapor adsorption kinetics on ACOF-1 upon exposure to I$_2$ vapor at 75 °C. **b** Static CH$_3$I adsorption kinetics on ACOF-1 upon exposure to CH$_3$I at 75 °C. **c** Exposure of saturated I$_2$/ACOF-1 and CH$_3$I/ACOF-1 to air at 25 °C for 7 days. **d** Experimental column breakthrough curves for I$_2$ vapor in an absorber bed packed with ACOF-1.

**e** Experimental column breakthrough curves for CH$_3$I in an absorber bed packed with ACOF-1. **f** Summary of dynamic I$_2$ and CH$_3$I adsorption capacities for ACOF-1 under various conditions. **g** Comparison of the I$_2$ and CH$_3$I dynamic capture capacities of ACOF-1 with reported adsorbents.

mesoporous (Fig. 3f). The Brunauer-Emmett-Teller (BET) specific surface areas determined for ACOF-1 and ACOF-1R were 1698 and 422.3 m$^2$/g, respectively. The total pore volumes are estimated to be 1.04 cm$^3$/g (ACOF-1) and 0.24 cm$^3$/g (ACOF-1R). The pore distribution in each COF was calculated using a non-local density functional theory (NLDFT) method, yielding average pore widths of 3.6 nm and 3.2 nm, respectively, consistent with the designed structure results (Fig. 3g). Thermal gravimetric analysis (TGA) revealed that ACOF-1 and ACOF-1R were stable up to ~340 °C and ~300 °C, respectively, under a N$_2$ atmosphere (Supplementary Fig. 9). PXRD further showed the frameworks of ACOF-1 and ACOF-1R could be maintained up to ~250 °C with heating (under vacuum) for 12 h (Supplementary Fig. 10). We next examined the chemical stability of ACOF-1 and ACOF-1R by soaking the COFs in cyclohexane, 12 M HNO$_3$, and 12 M NaOH solutions, respectively. PXRD and FT-IR data revealed that the frameworks retained their crystalline structures after three days of immersion under all these treatment conditions, implying excellent chemical stability

(Fig. 3h, i, Supplementary Fig. 11). The radiation resistance properties of ACOF-1 and ACOF-1R were evaluated by exposing samples to γ-ray irradiation. PXRD and FT-IR results confirmed the integrity of ACOF-1 and ACOF-1R after a 400 kGy γ-ray dose (Fig. 3h, i, Supplementary Fig. 11). Given the above structural and physical attributes, ACOF-1 offered many desirable advantages as an adsorbent for I$_2$ and CH$_3$I, whilst the cationic framework in ACOF-1R was expected to be suitable for I$_3^-$ capture. Therefore, we carried out I$_2$, CH$_3$I, and I$_3^-$ adsorption studies using ACOF-1 and ACOF-1R under various conditions to explore their practical utility as adsorbents for iodine pollutants.

**Iodine and iodomethane adsorption studies**

To investigate the I$_2$ and CH$_3$I capture ability of the COFs, ACOF-1 was initially selected for I$_2$ and CH$_3$I adsorption experiments in a static closed system at 75 °C (Supplementary Methods, Supplementary Fig. 12). As shown in Fig. 4a and Supplementary Fig. 13, ACOF-1 exhibited a progressive color change (darkening) upon exposure to I$_2$

vapor. ACOF-1 displayed very fast $I_2$ adsorption kinetics, with an uptake value reaching ~4.73 g/g of equilibrium capacity within 92 h (Fig. 4a). Subsequently, we carried out $CH_3I$ adsorption studies. As expected, very fast $CH_3I$ uptake kinetics and a high capacity of ~1.61 g/g of ACOF-1 were observed under 75 °C (Fig. 4b). These values are comparable to those of state-of-the-art sorbents under the same adsorption conditions, such as COF-TAPT[49] and SCU-COF-2[51] (Supplementary Tables 5, 6). Exposure of the $I_2$- and $CH_3I$-saturated COF (i.e., $I_2$/ACOF-1, $CH_3I$/ACOF-1) to air at 25 °C for 7 days resulted in negligible weight loss, suggesting a strong host-guest interaction between $I_2$ (or $CH_3I$) and ACOF-1 (Fig. 4c). Results indicate that ACOF-1 has great potential for the capture of $I_2$ and $CH_3I$ at low concentrations in gas streams. The fast kinetics and high $I_2$ and $CH_3I$ uptake performance of ACOF-1 prompted us to evaluate its dynamic adsorption ability, which was explored through fixed bed column breakthrough tests performed under various conditions. Breakthrough experiments for dynamic $I_2$ and $CH_3I$ capture under different conditions used ACOF-1 filled quartz columns (Supplementary Fig. 14). ACOF-1 exhibited fast $I_2$ uptake kinetics, reaching adsorption breakthrough-equilibrium within 67 h and 22 h at 25 °C and 75 °C, respectively, with the equilibrium dynamic capacities being 2.16 g/g and 0.41 g/g, respectively (Fig. 4d, f). To explore the effect of humidity on $I_2$ uptake, we next examined the dynamic $I_2$ adsorption performance of ACOF-1 using breakthrough experiments conducted at 50% relative humidity. Impressively, ACOF-1 effectively captured $I_2$ under 50% humidity conditions, reaching equilibrium after 56 h (~1.71 g/g) and 18 h (0.36 g/g) at 25 °C and 75 °C, respectively. The slightly lower uptake capacities of ACOF-1 at 50% relative humidity may have been due to some adsorption of $H_2O$ molecules at "multi-N nanotrap sites" during the dynamic breakthrough tests.

Further, we performed $CH_3I$ breakthrough experiments to assess the dynamic $CH_3I$ capture ability of ACOF-1. Figure 4e shows the concentration profile of $CH_3I$ exiting an adsorption bed packed with ACOF-1 as a function of time. The $CH_3I$ signals were detected at breakthrough times of 16 min and 12 min, with calculated adsorption capacities of 0.74 g/g at 25 °C and 0.39 g/g at 75 °C, respectively (Fig. 4e, f). For comparison, $CH_3I$ was detected after only 2 min in the absence of ACOF-1. The presence of water vapor typically strongly affects the $CH_3I$ adsorption performance of porous adsorbent materials. To test the effect of water vapor on the dynamic separation of $CH_3I$ by ACOF-1, we performed further $CH_3I$ capture experiments at 50% relative humidity. As expected, the presence of water vapor had negligible impact on the $CH_3I$ adsorption performance of ACOF-1 at 25 °C or 75 °C, suggesting suitability for practical applications. The calculated dynamic adsorption capacity reached as high as 0.50 g/g and 0.42 g/g at 25 °C and 75 °C, respectively (Fig. 4f). The dynamic $I_2$ and $CH_3I$ capture capacities of ACOF-1 surpassed the most sorbents reported to date (Fig. 4g, Supplementary Tables 7, 8). PXRD and SEM revealed that ACOF-1 retained its crystallinity and lychee-like morphology after the breakthrough studies (Supplementary Figs. 15, 16). The long breakthrough time, high dynamic capacity, radiation resistance, chemical stability, and reusability suggest that ACOF-1 is a very promising adsorbent for use in packed columns for $I_2$ and $CH_3I$ capture from off-gas streams.

## Triiodide adsorption studies
Due to the high environmental mobility of $I_3^-$, iodine-based contaminants can quickly move into groundwater, where they pose a serious threat to aquatic life and humans. Encouraged by the above results, we postulated that ACOF-1R containing pyridine-$N^+$ binding sites would offer abundant binding sites for $I_3^-$ anions. Sorption measurements were performed on ACOF-1R in $I_2$/NaI solutions ($I_2 + I^- \rightleftharpoons I_3^-$) under various conditions. Initial sorption experiments were performed to evaluate the working kinetics of ACOF-1R at an adsorbent to liquid ratio of 0.2 g/L in aqueous iodine solutions (Supplementary Fig. 17).

Aqueous iodine solutions contain $I_2$ and $I_3^-$, as evidenced by ultraviolet visible (UV-vis) spectroscopy (Supplementary Fig. 18). The initial experiments confirmed that ACOF-1R possessed rapid $I_3^-$ adsorption kinetics and a relatively slow $I_2$ capture ability in a saturated aqueous iodine solution, with an adsorption equilibrium time of ~120 min (Supplementary Fig. 18). Subsequently, the $I_3^-$ adsorption abilities of the ACOF-1R were determined in $I_2$/NaI ($I_2$: 50 ppm, NaI: 100 ppm) solutions (containing predominantly $I_3^-$ with a trace amount of $I_2$) and contaminated groundwater. ACOF-1R exhibited fast adsorption kinetics in the $I_2$/NaI solution, achieving equilibrium removal ratios > 93% and > 95% in 10 min and 60 min, respectively (Fig. 5a). Furthermore, ACOF-1R could remove over 99% of $I_3^-$ from contaminated groundwater ($I_2$: 50 ppm, NaI: 100 ppm) at an adsorbent to liquid ratio of 0.2 g/L with fast kinetics (Fig. 5b, c). The amounts of other anions, such as $NO_3^-$, $HCO_3^-$, $Cl^-$, $CH_3COO^-$ (Ac$^-$), and $SO_4^{2-}$, in contaminated groundwater are generally much higher than that of $I_3^-$. Thus, we next carried out experiments to assess the ion exchange selectively of the ACOF-1R towards $I_3^-$ in the presence of the other potentially competing anions. Figure 5d revealed the remarkably high selectivity of ACOF-1R towards $I_3^-$ over these other anions, suggesting a very strong affinity towards $I_3^-$ under high ionic strength conditions.

On the basis of fast kinetics and good selectivity of ACOF-1R towards triiodide, dynamic $I_3^-$ capture breakthrough experiments using $I_2$/NaI-spiked groundwater were conducted to simulate the treatment of contaminated water sources (Fig. 5e, Supplementary Fig. 19). Commercial activated carbon and different polymer adsorbents (such as D311, D101, D152, and IRA-402) were used for comparison. For ACOF-1R, $I_3^-$ broke through the adsorption bed after 423 mL, with a dynamic capacity of ~4.46 g/g at equilibrium (Fig. 5e). In comparison, all of the commercial sorbents showed poor dynamic $I_3^-$ capture performance under the same conditions. Durability and cycling stability are further important considerations for practical adsorbents. It is worth mentioning that the $I_3^-$ exchanged ACOF-1R packed column could readily be regenerated by elution with ethanol and saturated NaCl solution. The dynamic $I_3^-$ capture ability of ACOF-1R was almost unchanged after four cycles, suggesting suitability for practical applications (Fig. 5f). To the best of our knowledge, the dynamic $I_3^-$ capture capacity of ACOF-1R exceeded all adsorbents (under similar conditions) reported thus far (Fig. 5g, Supplementary Table 9). After the breakthrough experiments, the crystallinity and morphology of ACOF-1R were retained, indicating its excellent stability (Fig. 5h, Supplementary Fig. 20). The long breakthrough time and excellent durability suggest ACOF-1R is very effective for triiodide capture from contaminated solutions.

## Mechanism studies
The foregoing results demonstrate the excellent $I_2$, $CH_3I$, and $I_3^-$ capture performance of ACOF-1 and ACOF-1R under various conditions. Next, we carried out detailed characterization studies, including PXRD, FT-IR, XPS, Raman, and solid-state $^{13}C$ NMR, together with density functional theory (DFT) calculations to further investigate the adsorption mechanisms used by the COFs. After exposing ACOF-1 to $I_2$ vapor for 1 h, the PXRD peak at 2θ ~ 2.5° disappeared, with the other peaks at 2θ ~ 4.2°, 4.9°, 6.5°, and 8.4° disappearing after 3 h $I_2$ exposure. This suggested that $I_2$ gradually occupied the pore space in ACOF-1 (Supplementary Fig. 21). In the N 1s XPS spectra, the C-N = C, O = C-NH, C = N signals at 398.3, 399.3, and 400.0 eV in ACOF-1 shifted to 400.2, 401.2, and 402.1 eV, respectively, after $I_2$ exposure (Fig. 6a). This suggested the formation of charge-transfer complexes between $I_2$ and pyridine-N/hydrazine N sites in ACOF-1. Moreover, the FT-IR spectrum of ACOF-1 after saturated adsorption of $I_2$ showed a reduced intensity of the pyridine-N signal at 1537 cm$^{-1}$, further suggesting the strong intermolecular binding affinities between the $I_2$ and pyridine-N groups in the COF (Fig. 6b). ACOF-1 could readily be regenerated by soaking the $I_2$-saturated sample in ethanol to release the adsorbed iodine

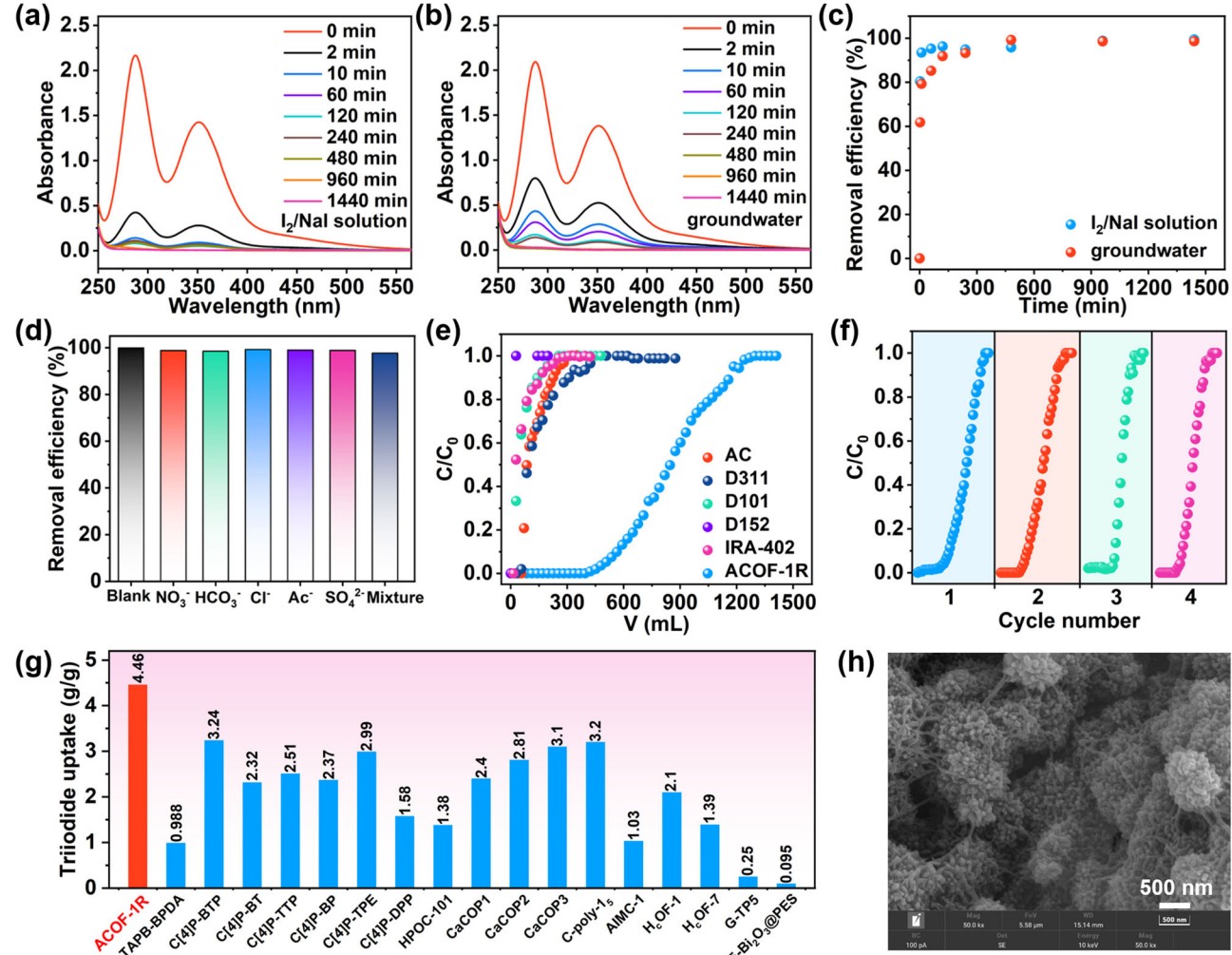

**Fig. 5 | Triiodide removal by ACOF-1R under different conditions. a** UV-vis spectra of $I_3^-$ adsorption for ACOF-1R in $I_2$/NaI solution. **b** UV-vis spectra of $I_3^-$ adsorption for ACOF-1R in $I_2$/NaI-spiked groundwater. **c** $I_3^-$ adsorption kinetics on ACOF-1R in $I_2$/NaI solution and $I_2$/NaI-spiked groundwater. **d** Selectivity of ACOF-1R for different anions in aqueous solution. **e** Experimental column breakthrough curves for $I_3^-$ in an absorber bed packed with ACOF-1R and other commercial sorbents. **f** Recycle test data for $I_3^-$ removal from groundwater (an absorber bed packed with ACOF-1R). **g** Comparison of triiodide uptake amount by ACOF-1R and other reported materials. **h** SEM image of ACOF-1R after adsorption of $I_3^-$.

(Supplementary Fig. 21). Solid-state $^{13}C$ NMR revealed that the pyridine-N sites in ACOF-1 fully reacted with $CH_3I$ to form [pyridine-$N^+ - CH_3$]I$^-$ (~58 ppm by solid state $^{13}C$ NMR) during the adsorption experiments (Fig. 6c). This was further confirmed by XPS, with the appearance of pyridine-$N^+$ signals at 401.6 eV (Fig. 6a). FT-IR spectroscopy confirmed this reaction process, with a new peak emerging at 1630 cm$^{-1}$ through the methylation (Fig. 6b). The Raman spectrum of ACOF-1R following iodine adsorption experiments showed bands at 107, 146, and 223 cm$^{-1}$, which could readily be assigned to symmetric and asymmetric stretching modes of $I_3^-$, and a symmetric stretching mode of $I_2$, respectively (Supplementary Fig. 22). Similar results were obtained using UV-vis spectroscopy to monitor iodine adsorption in aqueous solutions (Supplementary Fig. 18). XPS spectrum further revealed the $N^+$-$CH_3$ signal was retained after adsorption experiments. (Fig. 6a). FT-IR spectroscopy showed that the $CF_3SO_3^-$ anions in ACOF-1R were exchanged by $I_3^-$, evidenced by the significantly reduced intensities of the $S = O$ peaks at 1011 cm$^{-1}$ and C-S at 640 cm$^{-1}$ after the sorption experiments (Fig. 6b).

Accurate determination of the binding sites for iodine species in the COF frameworks is of great significance for understanding the adsorption mechanism and future design of iodine capture and separation materials. Therefore, we next carried out DFT calculations

to explore iodine species captured by ACOF-1 and ACOF-1R. On the basis of the above FT-IR and XPS results, we first considered $I_2$ binding on the N-containing hydrazine and pyridine sites of ACOF-1. The structural models of $I_2$/hydrazine (or pyridine) sites were chosen to represent the 2D layered structure of antiparallel AA stacking ACOF-1 (Fig. 6d). The binding free energies of $I_2$/hydrazine sites and $I_2$/pyridine sites were calculated to be -0.369 eV and -0.906 eV, respectively (Fig. 6d). The higher binding free energy of $I_2$/pyridine suggests a stronger binding affinity compared to $I_2$/hydrazine, thus $I_2$ will adsorb closer to the pyridine sites than the nearby hydrazine sites. Furthermore, the calculated binding energies of $I_3^-$/"multi-N nanotrap" sites and $I_5^-$/"multi-N nanotrap" sites were -0.508 eV and -0.605 eV, respectively, suggesting very strong binding affinities of "multi-N nanotraps" for iodine anions (Fig. 6e). The aforementioned solid-state $^{13}C$ NMR studies revealed that $CH_3I$ adsorbed on the pyridine-N sites in ACOF-1. We thus calculated the relative free energy diagrams for $CH_3I$ adsorption. As shown in Fig. 6f, g, $CH_3I$ adsorbed molecularly on pyridine-N sites forming a transition state with an increase in free energy (2.0 kcal/mol), followed by a free energy decrease to -8.6 kcal/mol with the release of I$^-$, accounting for the superior adsorption performance of ACOF-1 for $CH_3I$ under various conditions. Electrostatic potential (ESP) distribution analysis showed the local

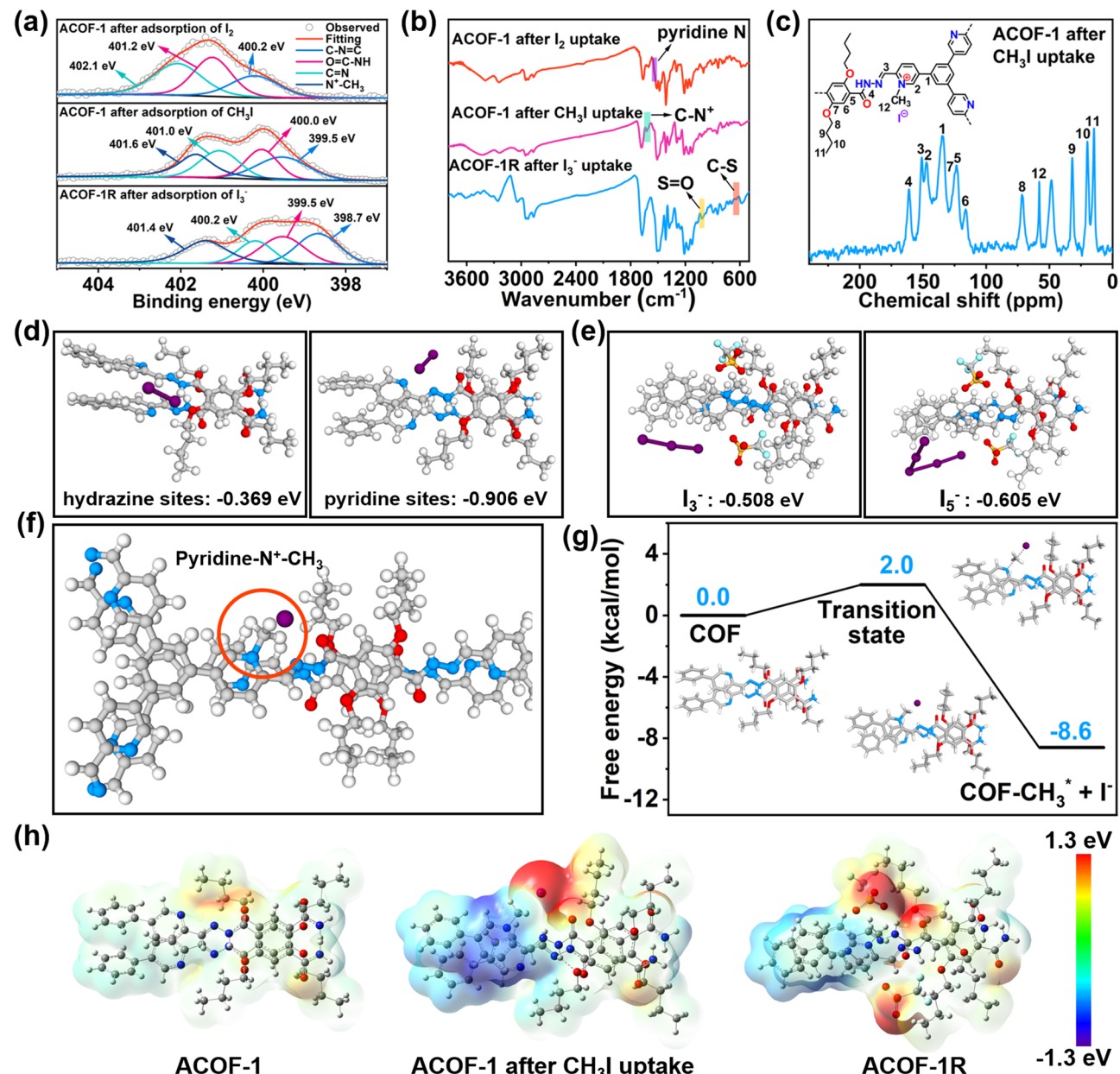

**Fig. 6 | I₂, CH₃I, and I₃⁻ adsorption mechanisms. a** N 1 s XPS spectra of ACOF-1 and ACOF-1R after adsorption of I₂, CH₃I, and I₃⁻. **b** FT-IR spectra of ACOF-1 and ACOF-1R after adsorption of I₂, CH₃I, and I₃⁻. **c** ¹³C CP-MAS solid-state NMR spectra of ACOF-1 after adsorption of CH₃I. **d** Structural models and binding energies of I₂/hydrazine and I₂/pyridine sites. **e** Structural models and binding energies of I₃⁻/pyridine and I₅⁻/pyridine sites. **f**, **g** Structural models of CH₃I/ACOF-1 and binding free energy diagrams from DFT calculations. **h** The electrostatic potential distributions for ACOF-1, ACOF-1 after CH₃I uptake, and ACOF-1R.

polarization of ACOF-1, ACOF-1 after adsorption of CH₃I, and ACOF-1R, revealing positive charges distributed close to the pyridine-N sites in the frameworks after the methylation reaction (Fig. 6h). This further explains the superior adsorption of CH₃I or I₃⁻ anions on pyridine-N (including pyridine-N⁺ sites) sites relative to other sites.

Taken together, the aforementioned experimental and theoretical results demonstrate that the developed covalent organic frameworks with antiparallel AA stacking structures are excellent adsorbents for iodine species under diverse conditions. The presence of antiparallel layers strongly influences the binding affinity between pyridine binding sites and iodine species. For instance, strong binding affinity for I₂ and CH₃I was realized through the cooperation of "multi-N nanotraps sites" composed of pyridine-N sites and hydrazine groups between layers, which were found to be more favorable for the adsorption of I₂ and CH₃I. Moreover, post-synthetic modification transformed the

neutral framework to a cationic framework ACOF-1R through a methylation reaction, enabling rapid triiodide (I₃⁻) uptake compared to other common anions, including NO₃⁻, HCO₃⁻, Cl⁻, Ac⁻, and SO₄²⁻ in the contaminated groundwater due to the high binding free energies at pyridine-N⁺ sites. In addition, the high porosity of the COFs allowed efficient mass transport, resulting in efficient dynamic I₂ and CH₃I capture, as well as I₃⁻ removal from polluted water. Further, the relationship between the COF structures and their adsorption performance has been clearly identified at a molecular level. We anticipate that our proof-of-concept study will promote the synthesis of improved COF-based adsorbents for iodine species and other target applications.

In summary, an antiparallel stacked COF (ACOF-1) and its cationic derivate (ACOF-1R) were synthesized and structurally characterized. Owing to its unique structural features, ACOF-1 showed a preferential

affinity toward $I_2$ and $CH_3I$, particularly for $CH_3I$, utilizing pyridine-N and hydrazine units (i.e., "multi-N nanotrap sites") that line the pores to strongly bind these molecules. ACOF-1 can dynamically capture $I_2$ and $CH_3I$ under various conditions. Notably, ACOF-1 showed a high dynamic $CH_3I$ capture capacity of ~0.74 g/g at 25 °C, surpassing the performance of most of the other reported COF-based adsorbents under similar conditions. It also retains excellent adsorption performance in the presence of water vapor. The cationic framework ACOF-1R can efficiently remove triiodide ions from aqueous solutions, resulting in a high uptake performance (~4.46 g/g) in contaminated groundwater. The high uptake and rapid adsorption kinetics demonstrated by ACOF-1 and ACOF-1R toward different iodine species were intrinsically related to the COF's antiparallel AA stacking mode. Such antiparallel AA stack structures imparted strong binding affinities with the various iodine species through the synergistic interaction between layers, thus enabling remarkable adsorption performances. This proof-of-concept work guides the development other of improved COF-based sorbents exploiting synergistic coordinative functionalities for iodine pollutants capture and other applications.

## Methods

### Materials and measurements

2,5-Dibutoxyterephthalohydrazide (DBTH, 99%) and 5,5',5"-(benzene-1,3,5-triyl)tripicolinaldehyde (BTPA, 98%) were purchased from Jilin Chinese Academy of Sciences-Yanshen Technology Co., Ltd. Ethanol (99.7%), Methanol (99.5%), 1,4-Dioxane (99%), Acetic acid (99.5%), N,N-Dimethylformamide (99.5%), 1,3,5-Trimethylbenzene (99%), Cyclohexane (99.7%), Methyl trifluoromethanesulfonate (97%), D311 macroporous weak alkaline acrylic anion exchange resin, Macropore adsorptive resin D101, D152 macroporous acrylic weak acid cation exchange resin, Amberlite IRA402, and Activated carbon were purchased from Shanghai Macklin Biochemical Co., Ltd (Shanghai, China). Ultrapure water was obtained from a Millipore system (18.25 MΩ·cm). All chemicals were sourced from commercial suppliers and used without further purification. The groundwater was collected in Mentougou, Beijing, China. Powder X-ray diffraction (PXRD) patterns were collected on a Rigaku SmartLab SE X-ray diffractometer equipped with a Cu Kα source. Fourier transform infrared spectra (FT-IR) were recorded on a SHIMADZU IRTracer-100. Scanning electron microscopy (SEM) images were recorded on TESCAN MIRA4 and TESCAN MIRA LMS Scanning Electron Microscopes. Transmission electron microscopy (TEM) images were obtained on a JEM-ARM200F electron microscope operating at 200 kV. High-resolution TEM images were obtained on a Thermo Fisher Scientific Titan Themis Z transmission electron microscope operating at 60 kV and equipped with a spherical aberration corrector. Solid-state $^{13}C$ CP/MAS NMR spectra were recorded on a Bruker AVANCE III HD 600 WB spectrometer with a 4.0 mm MAS probe and spin rate of 10 kHz. X-ray photoelectron spectroscopy (XPS) analyses were performed using a Thermo Scientific ESCALAB 250Xi spectrometer, equipped with a monochromatic Al Kα X-ray source. Thermogravimetric analyses (TGA) were performed on TESCAN MIRA LMS under a flow of nitrogen by heating samples from room temperature to 800 °C at a rate of 10 °C/min. BET surface areas were determined from $N_2$ adsorption/desorption isotherms collected at 77 K on a Micromeritics ASAP 2020 volumetric adsorption analyzer. Elemental analysis was performed on an Elemental Vario EL cube. Raman spectra were obtained from powder samples on a Jobin Yvon HR-800 Raman spectrometer equipped with a cobalt samba single-mode 514 nm diode laser. UV-vis spectra were recorded on a SHIMADZU UV-2700 UV-Vis spectrophotometer. UV-vis single point data were recorded on a Techcomp S1020 UV-Vis spectrophotometer. Dynamic $I_2$ and $CH_3I$ adsorption experiments were conducted on a custom-built breakthrough system. $CH_3I$ in the effluent from the adsorbent column was quantified using a gas chromatograph (Shimadzu GC2030) equipped with a flame ionization detector and GC column (SH-1, 0.25 mm × 0.25 μm × 30 m). Small and wide-angle X-ray scattering patterns were performed using a Xenocs Xeuss 3.0 equipped with a Cu microfocus sealed tube (30 W/30 μm) and generated at 50 kV and 0.6 mA. Dried sample powders were encapsulated with amorphous tape, then fixed to the sample holder using another piece of tape. The scattering patterns were recorded using Eiger2R 1 M detectors for the SAXS and WAXS measurement, $q = 4\pi sin\theta/\lambda$. Fit2D software was used for the data reduction with normalized circle gathering.

### Synthesis of ACOF-1

In a typical synthesis, 5,5',5"-(benzene-1,3,5-triyl)tripicolinaldehyde (BTPA, 39.3 mg) and 2,5-dibutoxyterephthalohydrazide (DBTH, 33.8 mg, 0.1 mmol) were dissolved in a mixed solvent solution containing 1,4-Dioxane (1 mL)/1,3,5-Trimethylbenzene (1 mL)/Acetic acid (6 M, 0.2 mL) in a 5 mL glass tube. Next, the mixture was sonicated and frozen in a liquid nitrogen bath and sealed with a gas torch. The tube was then heated at 120 °C for 3 d, after which the product was collected by filtration and washed several times with ethanol and acetone, yielding ACOF-1.

### Synthesis of ACOF-1R

100 mg of ACOF-1 was dispersed in dichloromethane (20 mL) by sonication for 10 min. Next, methyl trifluoromethanesulfonate (200 μL) was added to the ACOF-1 dispersion under constant stirring. After stirring for 24 h, the solid product was collected by centrifugation and washed several times with ethanol, yielding ACOF-1R.

### Dynamic triiodide ions capture from contaminated groundwater

Triiodide ion breakthrough experiments were conducted using a laboratory-scale fixed-bed reactor at 25 °C. 100 mg of ACOF-1R was packed into a quartz column (8 mm inside diameter, 2 mm wall thickness, 100 mm length) with degreased cotton filling the void space. Next, $I_2$/NaI-spiked groundwater ($I_2$: 50 ppm, NaI: 100 ppm) was flowed through the adsorbent column. The absorbance of the effluent at 288 nm ($I_3^-$) was measured at specific times. After the breakthrough experiments, ACOF-1R was washed with ethanol and then saturated in NaCl solution for 12 h, followed by washing with deionized water. The column was then used for the cycling test.

### Reporting summary

Further information on research design is available in the Nature Portfolio Reporting Summary linked to this article.

## Data availability

The authors declare that all the data supporting the findings of this study are available within the article (and Supplementary Information Files), or additional data are available from the corresponding author upon request. Source data are provided with this paper (https://doi.org/10.6084/m9.figshare.24579640).

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

## Acknowledgements

We gratefully acknowledge funding support from the National Science Foundation of China (Grants U2167218 (H.Y.); 22006036 (H.Y.); 22276054 (Z.C.)), the Beijing Outstanding Young Scientist Program (X.W.), and the Robert A. Welch Foundation (B-0027). G.I.N.W. is supported by a James Cook Research Fellowship from New Zealand Government funding, administered by the Royal Society Te Apārangi. We thank Dr. Jie Jin (Grant U2067215) with kindly discussion, Shiyanjia Lab (https://www.shiyanjia.com) for expert assistance with the DFT calculations.

## Author contributions

H.Y., X.W. and S.M. conceived and designed the research. Y.X., Q.R., X.L. and M.H. performed the synthesis and characterization. Y.X., Q.R., F.M., Y.W. and S.W. carried out the adsorption experiments. Y.X., X.L. and Z.C. performed and analyzed the TEM and DFT calculations. H.Y., X.W., G.I.N.W. and S.M. wrote the manuscript. All authors contributed to the discussion, and gave approval to the final version of the manuscript.

## Competing interests

The authors declare no competing interests.
