## [Peer Review File · Nature Communications]

REVIEWER COMMENTS

Reviewer #1 (Remarks to the Author):

In this manuscript, the authors present a strategy for converging cooperative functions into the nanospace of covalent organic frameworks for radioiodine capture from nuclear fuel waste and contaminated water sources. The structure and composition of ACOF-1 and derivative of ACOF-1R were characterized by FT-IR, NMR, XPS, PXRD, HRTEM, and N₂ sorption, etc., Experiment results disclosed showed excellent I₂ and CH₃I capture performance under dynamic adsorption conditions for ACOF-1, which can be also efficient for the removal of triiodide ions from aqueous solutions, resulting in a record-high uptake performance in contaminated groundwater. This work is of high importance for environmental radiochemistry. The manuscript is well written. Some revisions were suggested.

1. In general, two-dimensional COFs are usually eclipsed stacking, but the authors successfully prepared the antiparallel eclipsed stacking ACOF-1. I am wondering if the authors can explain as much as possible so that it can be of use to the broadest audience.

2. The authors claimed, "The sorption isotherms exhibited a type-IV adsorption profile..." However, the sorption isotherms look more like type II adsorption/desorption isotherms. The authors should carefully check the analysis results.

3. The dynamic I₃⁻ capture ability of ACOF-1R was almost unchanged after four cycles. In this reusability study, the authors do not define what was measured during the investigation. How did the authors determine reusability? Please make explanations.

4. In general, the dynamic CH₃I adsorption capacity of the adsorbent is lower than the static adsorption capacity. But why is the dynamic adsorption capacity of ACOF-1 for CH₃I higher than that of static adsorption?

5. Although this is a fundamental chemistry study related to the design of COF adsorbents for iodine-based pollutant capture, ACOF-1 and ACOF-1R demonstrated excellent dynamic adsorption performance under different conditions. It is suggested the authors calculate the cost of applying these COFs at scale.

Reviewer #2 (Remarks to the Author):

The presented paper focuses on preparation of the material with high binding affinities toward I₂, CH₃I, and I₃⁻. While the direction of preparation of materials capable capturing radioactive and volatile species is important, the type of materials and overall approach has been under exploration for a while. The motivation highlighted in the introduction does not sufficiently persuade the reviewer that the prepared material can effectively address the challenges outlined. The authors should provide a clearer explanation of how this specific material could potentially address the issue of nuclear waste pollution and how it might hypothetically be applied to challenges associated with nuclear waste. The material is purely organic, and it is unclear at which stage it can be effectively applied in the overall process of sequestration of radioactive species. Overall, the reviewer is not convinced that the introduction aligns well with the presented studies. This specific research direction has been explored not only with covalent organic frameworks (COFs) but also with examples involving cages and metal-organic frameworks (MOFs). In every study conducted on these porous materials, a slight improvement in the adsorption capacity of iodine species is usually reported. The reviewer finds it unclear how these studies fundamentally differ from the previous reports. What happens with this material at elevated temperatures? How does thermal and chemical resistance correlate or affect the COF stacking? The pXRD pattern (due to relatively broad peaks) does not allow for the judgment of subtle changes in structure stacking, which could be crucial for alterations in adsorption capacity. It is not clear how the findings in this paper can advance knowledge in this field to a level that distinguishes it from other reports in the same domain. Can the authors synthesize a similar material based on the principles reported in this paper, as per the reported mechanism, and demonstrate its functionality?

Reviewer #3 (Remarks to the Author):

In this work, the authors report on the design and synthesis of an antiparallel stacked covalent organic framework for the capture of I₂, CH₃I, and I₃⁻ under practical conditions. The authors investigate the structure-property relationship by performing various experiments to access the adsorption capacity for I₂, CH₃I, and I₃⁻ capture of the developed ACOF-1 and ACOF-1R. For example, the functionalized ACOF-1R possesses efficient decontaminate I₃⁻ polluted groundwater to drinking levels with a record-high uptake capacity of ~4.46 g/g established through column breakthrough tests. This work is highly interesting since it adds significant value to the recent strategy for dynamically capturing iodine pollutants, thus imparting advanced properties to the adsorbents. Taken together, this work has been conducted to a high standard, with appropriate techniques for the design of COFs for target applications, and the results are clear and sound. The manuscript is well-written and accessible, and the relevant results are presented in a clear manner. I believe that the manuscript merits publication in Nature Communications, subjected to several minor revisions though, as shown below:

(1) The authors claimed that the pyridine-N sites converted to cationic units by post-synthetic methylation, but why the hydrazide N was not methylated?

(2) Does the pore size distribution of ACOF-1R match its theoretical results? Have the authors considered the existence of anions in the pores?

(3) The authors employed methylation to convert the pyridine-N to the pyridine-N⁺ group, how did the author confirm the ratio conversion of the pyridine-N? Did the authors try to synthesize ACOF-1R by the one-pot approach instead of post-synthetic methylation?

(4) The authors should unify the format of the references. Such as reference 5 seems to lose the volume.

(5) The authors missed critical literature for iodine adsorptions, please update the references in the introduction section and supplementary Table 4. J. Am. Chem. Soc. 145, 9679-9685 (2023); J. Am. Chem. Soc. 145, 2544-2552 (2023).

We greatly appreciate the very positive comments and constructive suggestions from all reviewers, and we have revised the manuscript accordingly, as detailed in the responses below. The corresponding changes have been highlighted in yellow in the main text and Supplementary Information.

Reviewer #1:

In this manuscript, the authors present a strategy for converging cooperative functions into the nanospace of covalent organic frameworks for radioiodine capture from nuclear fuel waste and contaminated water sources. The structure and composition of ACOF-1 and derivative of ACOF-1R were characterized by FT-IR, NMR, XPS, PXRD, HRTEM, and N₂ sorption, etc., Experiment results disclosed showed excellent I₂ and CH₃I capture performance under dynamic adsorption conditions for ACOF-1, which can be also efficient for the removal of triiodide ions from aqueous solutions, resulting in a record-high uptake performance in contaminated groundwater. This work is of high importance for environmental radiochemistry. The manuscript is well written. Some revisions were suggested

Response: We are grateful to the reviewer for taking the time to evaluate our work and greatly appreciate the positive comments and support of our work.

Comment 1: In general, two-dimensional COFs are usually eclipsed stacking, but the authors successfully prepared the antiparallel eclipsed stacking ACOF-1. I am wondering if the authors can explain as much as possible so that it can be of use to the broadest audience.

Response: We appreciate the comment from the reviewer. In this work, we selected the DBTH linker for the COF synthesis due to the intramolecular hydrogen bonding involving the BHTH linker (N-H...O), which plays a decisive role in the orientation of the hydrazine groups on the linkers, thus yields ACOF-1 crystallized in the antiparallel AA stacking mode. Moreover, the dipole moment of C(δ^+) = O(δ^-) and O(δ^-) = R(δ^+) between layers can induce the adjacent layers to better crystallize by reverse parallel stacking.

Comment 2: The authors claimed, "The sorption isotherms exhibited a type-IV adsorption profile..." However, the sorption isotherms look more like type II adsorption/desorption isotherms. The authors should carefully check the analysis results.

Response: We appreciate the comment from the reviewer. We have corrected the type IV adsorption profile to type II sorption isotherms for ACOF-1 in the revised manuscript.

Comment 3: The dynamic I₃⁻ capture ability of ACOF-1R was almost unchanged after four cycles. In this reusability study, the authors do not define what was measured during the investigation. How did the authors determine reusability? Please make explanations.

Response: We appreciate the comment from the reviewer. We have added the reusability of ACOF-1 in the revised Supplementary Information (After I₃⁻ adsorption experiments, ACOF-1R was immersed in ethanol (30 mL). The solvent was exchanged every 1 h until the solution remained colorless. The solid COF was then dispersed in 40 mL of saturated NaCl overnight. The solid COF

was then filtered and washed several times with deionized water. Then, the ACOF-1R was subsequently returned to a I_3^- solution for further adsorption tests.).

Comment 4: In general, the dynamic CH_3I adsorption capacity of the adsorbent is lower than the static adsorption capacity. But why is the dynamic adsorption capacity of ACOF-1 for CH_3I higher than that of static adsorption?

Response: We appreciate the comment from the reviewer. We carried out CH_3I sorption experiments under various conditions (e.g. 25 °C, 75 °C). The static adsorption capacity of ACOF-1 was 1.61 g/g, while the dynamic adsorption capacity was 0.39 g/g at 75 °C. The CH_3I was fully adsorbed by ACOF-1 under breakthrough experimental conditions due to the slow went through the ACOF-1 packed column. The adsorption of CH_3I is a spontaneous exothermic reaction revealed by DFT calculations, which helps explain that the adsorption of CH_3I by ACOF-1 at 25 °C (0.74 g/g) is higher than at 75 °C.

Comment 5: Although this is a fundamental chemistry study related to the design of COF adsorbents for iodine-based pollutant capture, ACOF-1 and ACOF-1R demonstrated excellent dynamic adsorption performance under different conditions. It is suggested the authors calculate the cost of applying these COFs at scale.

Response: We appreciate the comment from the reviewer. We estimated the cost of synthesizing ACOF-1 to be ~\$2.84 USD/g, suggesting the economic feasibility of the adsorbent for iodine species capture.

Reviewer #2:

The presented paper focuses on preparation of the material with high binding affinities toward I_2 , CH_3I , and I_3^- . While the direction of preparation of materials capable capturing radioactive and volatile species is important, the type of materials and overall approach has been under exploration for a while. The motivation highlighted in the introduction does not sufficiently persuade the reviewer that the prepared material can effectively address the challenges outlined.

Response: We are grateful to the reviewer for taking the time to evaluate our work. We have reorganized the introduction section to highlight the material can effectively address the challenges outlined in the revised manuscript.

Comment 1: The authors should provide a clearer explanation of how this specific material could potentially address the issue of nuclear waste pollution and how it might hypothetically be applied to challenges associated with nuclear waste.

Response: We appreciate the comment from the reviewer. We have reorganized the introduction section to provide a more straightforward explanation of how this specific material could address the issue of nuclear waste pollution and how it can hypothetically be applied to challenges associated with nuclear waste in the revised manuscript.

We have also cited relevant references in the revised manuscript (Yoshida, N. & Kanda, J. Tracking the Fukushima radionuclides. *Science* **336**, 6085 (2012); Jia, T., Shi, K., Wang, Y., Yang, J. & Hou, X. Sequential separation of iodine species in nitric acid media for speciation analysis of ^{129}I in a PUREX

process of spent nuclear fuel reprocessing. *Anal. Chem.* **94**, 10959-10966 (2022); Zhang, X., Maddock, J., Nenoff, T. M., Denecke, M. A., Yang, S. & Schroder, M. Adsorption of iodine in metal-organic framework materials. *Chem. Soc. Rev.* **51**, 3243-3262 (2022); Liu, T., Zhao, Y., Song, M., Pang, X., Shi, X., Jia, J., Chi, L. & Lu, G. Ordered macro-microporous single crystals of covalent organic frameworks with efficient sorption of iodine. *J. Am. Chem. Soc.* **145**, 2544-2552 (2023); Liu, Y., Li, J., Lv, J., Wang, Z., Suo, J., Ren, J., Liu, J., Liu, D., Wang, Y., Valtchev, V., Qiu, S., Zhang, D. & Fang, Q. Topological isomerism in three-dimensional covalent organic frameworks. *J. Am. Chem. Soc.* **145**, 9679-9685 (2023).

Comment 2: The material is purely organic, and it is unclear at which stage it can be effectively applied in the overall process of sequestration of radioactive species. Overall, the reviewer is not convinced that the introduction aligns well with the presented studies.

Response: We appreciate the comment from the reviewer. In the Introduction of the revised manuscript, we have expanded the discussion to include potential applications of COFs in the overall process of capture of radioactive iodine species as part of nuclear fuel reprocessing and environmental remediation (We hypothesized that by converging cooperative functions into the nanospace of covalent organic frameworks (COFs), it should be possible to achieve control over the recognition and dynamic uptake efficiency of different iodine species. It would be ideal to capture of the I₂ and CH₃I at the first stage of the spent fuel reprocessing process (spent fuel dissolved in nitric acid solutions), as well as I₃⁻ capture from wastewater containing radioactive iodine ions.).

We have also cited relevant references in the revised manuscript (Yoshida, N. & Kanda, J. Tracking the Fukushima radionuclides. *Science* **336**, 6085 (2012); Jia, T., Shi, K., Wang, Y., Yang, J. & Hou, X. Sequential separation of iodine species in nitric acid media for speciation analysis of ¹²⁹I in a PUREX process of spent nuclear fuel reprocessing. *Anal. Chem.* **94**, 10959-10966 (2022); Zhang, X., Maddock, J., Nenoff, T. M., Denecke, M. A., Yang, S. & Schroder, M. Adsorption of iodine in metal-organic framework materials. *Chem. Soc. Rev.* **51**, 3243-3262 (2022); Liu, T., Zhao, Y., Song, M., Pang, X., Shi, X., Jia, J., Chi, L. & Lu, G. Ordered macro-microporous single crystals of covalent organic frameworks with efficient sorption of iodine. *J. Am. Chem. Soc.* **145**, 2544-2552 (2023); Liu, Y., Li, J., Lv, J., Wang, Z., Suo, J., Ren, J., Liu, J., Liu, D., Wang, Y., Valtchev, V., Qiu, S., Zhang, D. & Fang, Q. Topological isomerism in three-dimensional covalent organic frameworks. *J. Am. Chem. Soc.* **145**, 9679-9685 (2023).

Comment 3: This specific research direction has been explored not only with covalent organic frameworks (COFs) but also with examples involving cages and metal-organic frameworks (MOFs). In every study conducted on these porous materials, a slight improvement in the adsorption capacity of iodine species is usually reported. The reviewer finds it unclear how these studies fundamentally differ from the previous reports.

Response: We appreciate the comment from the reviewer. In our work, we prepared COFs with antiparallel AA stacked structures which showed excellent uptake of I₂ and CH₃I molecules and I₃⁻ ions under diverse conditions. The presence of antiparallel layers strongly enhanced the binding affinity between pyridine binding sites and iodine species. For instance, strong binding affinity for I₂ and CH₃I was realized through the cooperation of “multi-N nanotrap sites” composed of pyridine-N

sites and hydrazine groups between layers in a neutral COF (ACOF-1), resulting in strong adsorption of I₂ and CH₃I. Moreover, post-synthetic methylation transformed the neutral framework to a cationic framework (ACOF-1R), which showed selective triiodide uptake compared to other common anions, including NO₃⁻, HCO₃⁻, Cl⁻, CH₃COO⁻, and SO₄²⁻ in the contaminated groundwater. The high selectivity towards I₃⁻ related to the high binding free energy of I₃⁻ at pyridine-N⁺ sites. In addition, the high porosity of the COFs ensured efficient mass transport, resulting in fast dynamic I₂ and CH₃I capture from the gas phase, as well as I₃⁻ removal from polluted water. The relationship between the COF structures and their adsorption performance was probed at a molecular level using spectroscopic studies and DFT calculations. To our knowledge, this is the first systematic investigation of antiparallel stacked layered COFs with three-dimensional “multi-N nanotrap sites” as I₂, CH₃I, and I₃⁻ adsorbents. Our results demonstrate that binding affinity between adsorbent and iodine species is the most critical determinant, together with the spatial environment and exposed binding sites in the adsorbents.

Comment 4: What happens with this material at elevated temperatures? How does thermal and chemical resistance correlate or affect the COF stacking? The PXRD pattern (due to relatively broad peaks) does not allow for the judgment of subtle changes in structure stacking, which could be crucial for alterations in adsorption capacity.

Response: We appreciate the comment from the reviewer. We carried out PXRD measurements of ACOF-1 after heating the samples at different temperatures for 12 h (under vacuum). The results showed that the frameworks were retained up to 250 °C, suggesting potential applications under particle capture under practical I₂ and CH₃I capture conditions (the practical industrial radioactive molecular iodine treatment conditions of temperature usually at 25~150 °C during the reprocessing of nuclear fuel rods). For structure stacking judgment, we carried out small and wide-angle X-ray scattering (SAXS/WAXS) measurements to corroborate the successful synthesis of ACOF-1 and ACOF-1R. The (100), (110), (200), and (210) planes of ACOF-1 were observed by 2D SAXS image and pattern, consistent with the PXRD and simulated antiparallel AA stacking results. ACOF-1R showed similar results (both before and after methylation, $q = 0.17 \text{ \AA}^{-1}$ in the SAXS patterns), suggesting that the crystallinity of the COF was retained after the methylation reaction. ACOF-1R maintained P6cc space group, with a slight unit cell parameter change of in $a = 42.14 \text{ \AA}$ and $c = 7.38 \text{ \AA}$, compared with the ACOF-1 ($a = 42.31 \text{ \AA}$, $c = 7.23 \text{ \AA}$). Moreover, the intensity of (100), (110), (200), and (210) reflections in ACOF-1R (from $q = 0.2\text{-}2.0 \text{ \AA}^{-1}$) were slightly changed compared with the ACOF-1, suggesting the pore environment changed after methylation. In addition, we selected the DBTH linker for the COF synthesis due to the intramolecular hydrogen bonding involving the DBTH linker (N-H...O), which plays a pivotal role in orienting the hydrazine groups on the linkers, resulting in ACOF-1 crystallizing in the antiparallel AA stacking mode. This was further revealed by comparison of the charge density map and interlayer differential charge density of antiparallel AA stacking and eclipsed AA stacking structures. The electrostatic repulsion of alkyl oxygen/hydrazine groups increases the interlayer distance and decreases the interlayer interaction in the eclipsed AA stacking structure. However, the antiparallel AA stacking mode avoids charge repulsion due to intra- and interlayer hydrogen bonds and electrostatic interactions, thus promoting the formation of the Antiparallel AA stacking structures. Taken together, we concluded that the ACOF-1 matches the structural model in Fig. 1b. Moreover, the dipole moment of $C(\delta^+) = O(\delta^-)$ and $O(\delta^-) = R(\delta^+)$ between

layers can induce the adjacent layers to better crystallize by reverse parallel stacking. Please see below results:

PXRD patterns of ACOF-1 and ACOF-1R after heating at different temperatures for 12 h (under vacuum).

2D SAXS image and pattern of ACOF-1.

2D SAXS image and pattern of ACOF-1R.

2D WAXS image (left) and pattern (right) of ACOF-1.

2D WAXS image (left) and pattern (right) of ACOF-1R.

Antiparallel AA stacking

Eclipsed AA stacking

Total charge density of ACOF-1 with the structural model assuming the antiparallel AA and eclipsed AA stacking configurations

Antiparallel AA stacking

Eclipsed AA stacking

Interlayer differential charge density of ACOF-1 with the structural model assuming the antiparallel AA and eclipsed AA stacking configurations

Total charge density of ACOF-1 with the structural model assuming the antiparallel AA and eclipsed AA stacking configurations in an iso-surface of 0.13 e per Å³ (top). Interlayer differential charge density of ACOF-1 with the structural model assuming the antiparallel AA and eclipsed AA stacking

configurations in an isosurface of 5.7×10^{-4} e per \AA^3 (bottom). Pink and blue regions represent decreased or increased charge density, respectively.

Total energy of DFT-optimized ACOF-1 in the antiparallel AA and eclipsed AA unit cell.

Stacking model	Interlayer π - π distance (\AA)	Total energy (eV)
Antiparallel	3.62	-2489.21
Eclipsed	3.96	-2487.43

Comment 5: It is not clear how the findings in this paper can advance knowledge in this field to a level that distinguishes it from other reports in the same domain. Can the authors synthesize a similar material based on the principles reported in this paper, as per the reported mechanism, and demonstrate its functionality?

Response: We appreciate the comment from the reviewer. We have made changes to the manuscript, particularly in the introduction section, to make it more straightforward that this work is innovative. Among the potentially troublesome radionuclides are the isotopes of iodine that account for 0.69% of the ^{235}U slow-neutron fission products, including short-lived ^{131}I and long-lived ^{129}I , which typically exist as I_2 and CH_3I in air, or molecular/ionic species in water ($\text{I}_2 + \text{I}^- \rightleftharpoons \text{I}_3^-$). Due to the highly volatile nature of I_2 and CH_3I , most of which are released into the air during the acid dissolution of the spent fuel reprocessing process. Another fraction exist as molecular/ionic species dissolved in acidic nuclear waste. Due to their radiotoxicity, volatility at relatively lower temperatures, and mobility, radioactive I_2 , CH_3I , and I_3^- can quickly spread into the environment, impacting living organisms and inducing disease. In addition, nuclear accidents caused by human activity can also lead to the release of large amounts of radioactive iodine. Effective radioactive iodine management, particularly at the first stage of the spent fuel reprocessing process, is essential for the safe disposal of radioactive waste from power generation and the prevention of off-site migration of contaminated aqueous solutions. However, radioactive iodine management and environmental remediation are technically very challenging, owing to the complexity of the systems harboring the iodine species.

To address the challenge of radioiodine capture, researchers seek adsorbents that can selectively capture iodine species to allow long-term storage and mitigate environmental risks. However, inorganic materials demonstrated relatively low capacity towards I_2 , CH_3I , and I_3^- . Most of porous material spontaneously release the adsorbed iodine because of the weak interactions between the pore of framework and iodine molecules. Moreover, most of the adsorption studies reported to date have focused on the adsorptive-capture of I_2 vapor, CH_3I , and/or I_3^- in closed systems, with only a few works exploring dynamic iodine species removal. Moreover, since radioactive molecular iodine and organic iodides (e.g., CH_3I) coexist in off-gas streams and I_3^- dissolved in contaminated water, it is particularly important to develop adsorbents that can dynamically and efficiently capture these diverse iodine-containing molecules/or ions under practical conditions, motivating detailed investigations.

How to improve the host-guest interaction between porous materials and iodine species is the key to overcome the above challenges. In this work, we report robust covalent organic frameworks possessing unique antiparallel stacked structures (denoted as ACOF-1 and ACOF-1R), good radiation resistance, high chemical stability, and excellent selectivity towards I_2 , CH_3I , and I_3^- . Our COFs thus

meet the numerous technical challenges of removing I_2 , CH_3I , and I_3^- under various conditions, allowing the fast dynamic capture of I_2 vapor and CH_3I from off-gas streams and I_3^- dissolved in contaminated water sources. The structure-property relationships in a neutral framework (ACOF-1) and a cationic framework (ACOF-1R) that underpin their outstanding iodine-based adsorption performance were also explored and established. To our knowledge, this is the first systematic investigation of antiparallel stacked layered COFs with three-dimensional “multi-N nanotrap sites” as I_2 , CH_3I , and I_3^- adsorbents. Further, the strategy described herein to construct antiparallel stacked COFs could readily be applied for the task-specific design of functional adsorbent materials for other applications. For example, antiparallel laminated COFs have recently been synthesized in our laboratory. This will be reported in the near future.

We have also cited relevant references in the revised manuscript (Yoshida, N. & Kanda, J. Tracking the Fukushima radionuclides. *Science* **336**, 6085 (2012); Jia, T., Shi, K., Wang, Y., Yang, J. & Hou, X. Sequential separation of iodine species in nitric acid media for speciation analysis of ^{129}I in a PUREX process of spent nuclear fuel reprocessing. *Anal. Chem.* **94**, 10959-10966 (2022); Zhang, X., Maddock, J., Nenoff, T. M., Denecke, M. A., Yang, S. & Schroder, M. Adsorption of iodine in metal-organic framework materials. *Chem. Soc. Rev.* **51**, 3243-3262 (2022); Liu, T., Zhao, Y., Song, M., Pang, X., Shi, X., Jia, J., Chi, L. & Lu, G. Ordered macro-microporous single crystals of covalent organic frameworks with efficient sorption of iodine. *J. Am. Chem. Soc.* **145**, 2544-2552 (2023); Liu, Y., Li, J., Lv, J., Wang, Z., Suo, J., Ren, J., Liu, J., Liu, D., Wang, Y., Valtchev, V., Qiu, S., Zhang, D. & Fang, Q. Topological isomerism in three-dimensional covalent organic frameworks. *J. Am. Chem. Soc.* **145**, 9679-9685 (2023).

Reviewer #3:

In this work, the authors report on the design and synthesis of an antiparallel stacked covalent organic framework for the capture of I_2 , CH_3I , and I_3^- under practical conditions. The authors investigate the structure-property relationship by performing various experiments to access the adsorption capacity for I_2 , CH_3I , and I_3^- capture of the developed ACOF-1 and ACOF-1R. For example, the functionalized ACOF-1R possesses efficient decontaminate I_3^- polluted groundwater to drinking levels with a record-high uptake capacity of ~4.46 g/g established through column breakthrough tests. This work is highly interesting since it adds significant value to the recent strategy for dynamically capturing iodine pollutants, thus imparting advanced properties to the adsorbents. Taken together, this work has been conducted to a high standard, with appropriate techniques for the design of COFs for target applications, and the results are clear and sound. The manuscript is well-written and accessible, and the relevant results are presented in a clear manner. I believe that the manuscript merits publication in Nature Communications, subjected to several minor revisions though, as shown below:

Response: We are grateful to the reviewer for taking the time to evaluate our work and greatly appreciate the positive comments and support of our work.

Comment 1: The authors claimed that the pyridine-N sites converted to cationic units by post-synthetic methylation, but why the hydrazide N was not methylated?

Response: We appreciate this excellent suggestion. The methylation of different nitrogen sites can be regulated by controlling the reaction conditions and methylation reagents. In this work, methyl trifluoromethanesulfonate, as an electrophile, preferentially reacts with electron-rich pyridine nitrogen at room temperature. Cationic units can be generated in ACOF-1 pyridine-N sites. The electron donor energy of hydrazide N is relatively weak, and the hydrazide N methylation reaction usually needs to be carried out under heating conditions.

Comment 2: Does the pore size distribution of ACOF-1R match its theoretical results? Have the authors considered the existence of anions in the pores?

Response: We appreciate the constructive comment from the reviewer. The experimental PXRD pattern of ACOF-1R matched well with simulated results by considering CF_3SO_3^- anions located close to the pyridine- N^+ sites. Considering the existence of anions in the pores of ACOF-1R, the measured pore width was 3.2 nm by N_2 sorption isotherm at 77 K, consistent with the theoretical results.

Comment 3: The authors employed methylation to convert the pyridine-N to the pyridine- N^+ group, how did the author confirm the ratio conversion of the pyridine-N? Did the authors try to synthesize ACOF-1R by the one-pot approach instead of post-synthetic methylation?

Response: We appreciate the comment from the reviewer. The ratio conversion of the pyridine-N to the pyridine- N^+ group could be controlled by adding a specific amount of methylation reagent (methyl trifluoromethanesulfonate) in the reaction. The elemental analysis was used to determine the ratio conversion of the pyridine-N. The results showed that 90% of the pyridine-N converted to the pyridine- N^+ through methylation. We tried the one-pot approach to synthesize the COF-1R, but failed. Please see below PXRD and FT-IR results.

PXRD pattern and FT-IR spectrum of ACOF-1R by the one-pot approach.

Comment 4: The authors should unify the format of the references. Such as reference 5 seems to lose the volume.

Response: We appreciate the comment from the reviewer. We have added the volume to reference 5 (Hao, M., Liu, Y., Wu, W., Wang, S., Yang, X., Chen, Z., Tang, Z., Huang, Q., Wang, S., Yang, H. & Wang, X. Advanced porous adsorbents for radionuclides elimination. *EnergyChem* **5**, 100101 (2023).).

And we have went through the manuscript to unify the format of the references in the revised manuscript.

Comment 5: The authors missed critical literature for iodine adsorptions, please update the references in the introduction section and supplementary Table 4. *J. Am. Chem. Soc.* 145, 9679- 9685 (2023); *J. Am. Chem. Soc.* 145, 2544-2552 (2023).

Response: We appreciate the comment from the reviewer. We have cited these references in the revised manuscript (Liu, T., Zhao, Y., Song, M., Pang, X., Shi, X., Jia, J., Chi, L. & Lu, G. Ordered macro-microporous single crystals of covalent organic frameworks with efficient sorption of iodine. *J. Am. Chem. Soc.* **145**, 2544-2552 (2023); Liu, Y., Li, J., Lv, J., Wang, Z., Suo, J., Ren, J., Liu, J., Liu, D., Wang, Y., Valtchev, V., Qiu, S., Zhang, D. & Fang, Q. Topological isomerism in three- dimensional covalent organic frameworks. *J. Am. Chem. Soc.* **145**, 9679-9685 (2023)).

Again, we thank all reviewers for the constructive suggestions, which have made our manuscript much improved.

REVIEWERS' COMMENTS

Reviewer #1 (Remarks to the Author):

The author has made all the revision, according to the reviewers' comments. And these new results can well support their claim. Thus, I give acceptance recommendation for the current form.

Reviewer #3 (Remarks to the Author):

[Note from the Editor: Reviewer #3 was asked to look also over the response given to reviewer #2]

In my opinion, all of the comments have been well addressed. The revised manuscript in present version can be accepted.

REVIEWER COMMENTS

Reviewer #1:

The author has made all the revision, according to the reviewers' comments. And these new results can well support their claim. Thus, I give acceptance recommendation for the current form.

Response: We are grateful to the reviewer for taking the time to evaluate our work and greatly appreciate the positive comments and support of our work.

Reviewer #3:

[Note from the Editor: Reviewer #3 was asked to look also over the response given to reviewer #2]

In my opinion, all of the comments have been well addressed. The revised manuscript in present version can be accepted.

Response: We are grateful to the reviewer for taking the time to evaluate our work and greatly appreciate the positive comments and support of our work.